# An Adaptive Pedaling Assistive Device for Asymmetric Torque Assistant in Cycling

**DOI:** 10.3390/s23052846

**Published:** 2023-03-06

**Authors:** Jesse Lozinski, Seyed Hamidreza Heidary, Scott C. E. Brandon, Amin Komeili

**Affiliations:** 1School of Engineering, University of Guelph, Guelph, ON N1G 2W1, Canada; 2Department of Biomedical Engineering, University of Calgary, Calgary, AB T2N 1N4, Canada

**Keywords:** knee rehabilitation, cycling ergometer, kinematics, electric bicycle

## Abstract

Dynamic loads have short and long-term effects in the rehabilitation of lower limb joints. However, an effective exercise program for lower limb rehabilitation has been debated for a long time. Cycling ergometers were instrumented and used as a tool to mechanically load the lower limbs and track the joint mechano-physiological response in rehabilitation programs. Current cycling ergometers apply symmetrical loading to the limbs, which may not reflect the actual load-bearing capacity of each limb, as in Parkinson’s and Multiple Sclerosis diseases. Therefore, the present study aimed to develop a new cycling ergometer capable of applying asymmetric loads to the limbs and validate its function using human tests. The instrumented force sensor and crank position sensing system recorded the kinetics and kinematics of pedaling. This information was used to apply an asymmetric assistive torque only to the target leg using an electric motor. The performance of the proposed cycling ergometer was studied during a cycling task at three different intensities. It was shown that the proposed device reduced the pedaling force of the target leg by 19% to 40%, depending on the exercise intensity. This reduction in pedal force caused a significant reduction in the muscle activity of the target leg (*p* < 0.001), without affecting the muscle activity of the non-target leg. These results demonstrated that the proposed cycling ergometer device is capable of applying asymmetric loading to lower limbs, and thus has the potential to improve the outcome of exercise interventions in patients with asymmetric function in lower limbs.

## 1. Introduction

Exercise is one of the noninvasive interventions for treatment and pain management of patients with lower limb conditions. Although the risks of injury due to falling, joint sprain, and muscle strain are minimal in aquatic interventions [1,2,3,4], the majority of patients with lower limbs conditions, such as knee osteoarthritis (OA), spend most of their time on land, performing the activities of daily living, limiting the accessibility of water-based activities [5]. Studies showed that land-based aerobic/cardiovascular exercises, such as jogging and walking, have at least short-term, with potential long-term, beneficial treatment effects for knee OA pain management and physical function [6]. However, there are concerns regarding the increased risk of localized stresses in load-bearing joints during open kinetic-chain activities, such as jogging, particularly when sensory control of the lower limb is compromised, a common symptom in older adults with impaired joint proprioception [7]. Open kinetic-chain exercises are associated with a higher risk of injuries in the elderly [8,9,10,11], whose declined muscle coordination and joint stability due to neuromuscular dysfunction are linked to balance problems [12,13,14]. These physiological barriers associated with aging and neuromuscular complications may put the elderly and patients with knee OA in a dilemma to participate in activities for fear of repercussions following joint injuries. To address this concern, clinicians considered low-impact exercises that require less neuromuscular coordination for the elderly with joint diseases.

Supervised cycling exercise training using stationary bicycles, called cycling ergometers, has been used as a surrogate land-based exercise for cardiovascular disease [1], knee OA physiotherapy [4,15] stroke rehabilitation [16], and post-surgery strength recovery [15]. The musculoskeletal degrees of freedom during cycling is much more limited compared to other land-based activities such as walking and running, thus it is associated with less fall hazard and altered biomechanics, making it an ideal treatment for improving joint mobility in patients with prolonged musculoskeletal conditions in the lower limbs. For instance, a 12-week cycling exercise regimen including 20–30 min/day, 3 days/week at an exercise intensity of 40–50% heart rate reserve (HRR) reduced joint pain, stiffness, and physical limitations and increased muscle strength by approximately 30% [17]. The participants in this randomized study were a group of 48 older and middle-aged adults with inactive lifestyles [17], a common demographic of interest for this kind of study. Patients with knee OA who participated in low-intensity (40% HRR) and high-intensity (70% HRR) cycling exercise training experienced pain relief and an enhanced quality of life compared to the sedentary control [18,19], indicating that cycling is not only effective for joint function, but also has a meaningful impact on the social life of patients [20]. Despite the common use of aerobic exercises in patients with hip and knee OA, the exercise intensity, session frequency and duration, and program period are very heterogeneous [21]. With the current cycling rehabilitation devices, the clinician subjectively adjusts the intensity and duration of the cycling exercise for each patient considering the symptomatic pain based on experience rather than objective measures. A detailed description of these adjustments rarely exists, reducing the reproducibility of the intervention delivered in clinical trials. 

Therefore, the first gap in using conventional cycling ergometer devices is the lack of a system to collect, analyze, and monitor patient performance, giving the practitioner a tool for data-driven assessment of treatment effectiveness. To achieve this with conventional cycling ergometers, a secondary system for monitoring health and performance factors should be improvised during the exercise, which would require bulky instrumentation and on-site supervision [22]. For instance, the NOTTABIKE cycling ergometer [23] was a recumbent stationary bicycle instrumented with available torque, force, and crank position sensing system, that required wired connections to the data acquisition system. Another limitation of available cycling ergometers is related to the inability to train each leg with different intensities during the intervention program. The current cycle ergometers cannot adjust forces and moments independently for each leg. For instance, Active Pedal Exerciser (APE) [24] was developed for the rehabilitation of neuromotor injuries without differentiating the pedaling power capacity of each of the lower limbs. In another work, Abdar et al. [25,26] created a cycling ergometer to improve the motor function of patients with Parkinson’s disease. Their device had a PLC-driven servomotor that assists the rider in maintaining a particular cycling cadence by providing a uniform cycling power assistance that was adjusted every 20 s, and thus asymmetric assistance was not feasible with their technology. This is a serious problem for clinicians and patients when there is asymmetry function between two legs (e.g., knee replacement post-operation, stroke, acute injuries, and knee OA), but both legs are involved in the same exercise. In such a case, the injured/weaker limb could be overloaded when it is forced to meet the same demand set to the healthy leg, resulting in pain and fatigue development in the inhibited limb, which is associated with an increased risk of injury [17]. The other possibility is that the weaker leg could “hide” and do no work while the healthy leg compensates. Therefore, there is a need for a rehabilitation device that is equipped with a biofeedback system to collect biomechanical/physiological data and can monitor and adjust the pedaling power according to users’ torque generation capacity. 

The present work seeks to develop a novel rehabilitation tool and prove its potential in exercising legs with different intensities such that the injured leg (target limb) is burdened with less mechanical loads than the healthy leg during the pedaling task. Therefore, this project aimed at prototyping the Adaptive Pedalling Assistant Device (APAD), and showing that it can provide asymmetrically-assisted cycling torque without significantly affecting kinematics. It was hypothesized that APAD would maintain the kinematics of pedaling while reducing the muscle force of the targeted leg. To assess the study hypothesis, the kinematics of movement was analyzed using the motion capture system, while the effect of asymmetric assistive torque on the kinetics of lower limbs was quantified using the pedal force sensors and muscle electromyography (EMG). 

## 2. Materials and Methods

### 2.1. Instrumentation and Design

The central device of this project, APAD, was built off a Rad Mission™ electric bike (Rad Power Bikes Inc., Washington, WA, USA) equipped with a 450 W brushless direct current rear-hub motor and a 48 V battery. The original Rad Mission ebike had a throttle control to adjust the motor output power (Figure 1). In this study, a custom controller was implemented to interface the Rad Mission motor controller with custom pedal force and crank position sensing systems. 

The force transducer was based on strain gauges connected using a Wheatstone (half) bridge amplifier circuit [27] glued to the side and front face of the crank and soldered onto two separate protoboards with two 9-volt power sources. Force sensor data was transferred to the bike custom controller via an NRF24 radio module. The optimized location of strain gauges was obtained by conducting a finite element study of crank maximum deformation. The strain gauge circuit calibration was performed using static weights of up to 20 kg hung from the crank arm at horizontal and vertical positions to measure the crank perpendicular and axial forces, respectively.

As shown in Figure 1, the crank position sensing system had an ‘array’ of 36 hall sensors (A3144E) arranged in a circular 3D-printed hub installed coaxially with the crank bottom bracket, giving a resolution of 10 degrees for identifying the angular position of the crank. The hall sensors each had a fixed address regarding their physical placement on the fixture and, accordingly, to the GPIO pin on the APAD custom controller. A magnet was attached to the back of the crank arm to trip hall sensors. The coordinate system identifying the crank orientation was defined such that the top dead center (TDC) and bottom dead center (BDC) of the pedal stroke represented 90° and 270°, respectively. Accordingly, a horizontal crank during the downstroke represented 180°. In the present work, the APAD hub motor was instructed to turn ON when the target leg (right leg in our experiments with healthy participants) was at TDC, and OFF when the target leg reached the BDC. In other words, this scheme split up a full revolution of the crank into “ON” 90° ≤ crank angle < 270° and “OFF” phases, 270° ≤ crank angle < (360 + 90)°. When the crank tracker detected a transition from the OFF to the ON phase (i.e., TDC), an open-loop feedback “ramp” function of 0.3 s was used to help energize the motor and avoid input delay. In Table 1, the main components of the APAD prototype are presented. 

The logic flow of the motor control program is described in Figure 2. In this application, an Arduino Mega was used as the main controller performing two main tasks; (1) after pressing the start button and adjusting the assistive torque level, the controller turns on the motor and activates the electronic power converter through which the force sensor and crank position sensing system are initialized and communicate with the controller, and (2) the control method, shown in Figure 3, is executed to control the assistive torque. When the crank position sensing system identifies the crank in the downstroke phase (crank angle 90° to 270°), the controller turns on the BLDC motor. When the crank was at the TDC position (crank angle 90°), the motor power was set at its maximum to eliminate the time lag when increasing it to the set value, and then, the motor power decreased linearly to the set power in 50 ms based on a ramp function.

### 2.2. Experimental Protocol

#### 2.2.1. Motion Capture

Bilateral lower-limb joint angles were recorded using a Vicon 9-cameras capture motion system at 100 Hz (Vicon Motion Systems Ltd., Yarnton, UK). 24 retroreflective motion capture markers were used to track left and right lower-limb and crank kinematics. There were three cluster markers on each thigh and shank, as well as a marker on each leg to indicate the hip, knee, and ankle joints as per Vicon (Yarnton, UK) documentation [28]. Markers were placed on the crank at the bottom bracket ends, midway on the crank arm, and the edges of the pedals (Figure 1). All tests were conducted in the Biomechanics Lab at the University of Guelph. 

#### 2.2.2. EMG Data 

Four EMG sensors (Trigno™, Delsys Inc., Natick, MA, USA) were placed on the legs; over the left and right vastus-lateralis-quadricep (VL) muscles in the same orientation as that of Zebis et al. [29], as well as the left and right gastrocnemius-lateralis (GL) calf muscle [30]; as per the guidelines in [31]. The EMG sensor locations were shaved and cleaned with alcohol pads to minimize EMG signal noises. EMG data were sampled at 2000 Hz (Vicon), then processed in MATLAB (The Mathworks, Natick, MA, USA) using a 4th order Butterworth bandpass filter (20–350 Hz), full-wave rectification, and a moving average (50 ms window). Enveloped EMG were downsampled to match kinematic data (100 Hz), and magnitude-normalized to each subject’s maximum recorded EMG for each muscle across all experimental trials. Thus, EMG data ranged from 0 to 1. Finally, the root-mean-square (RMS) of the filtered data was taken to find the moving average and applied across a window of 50 ms [32]. 

#### 2.2.3. Kinematics of Motion

The crank orientation was calculated once from the motion capture system using the markers attached to the cranks, pedals, and the proposed crank position sensing system, and results were compared to validate the performance of the crank position sensing system. Amongst kinematics parameters measured using the motion capture system, the knee joint angle and knee lateral deviations are more relevant to the present study. The knee joint angle was calculated by measuring the angle between the line segments representing the femur and tibia; the femur line connected the lateral knee (epicondyle) with the greater trochanter of the femur, and the tibia line connected the lateral ankle (malleolus) with the lateral knee (epicondyle). The lateral-medial movement of the knee joint was identified by the trace of the marker attached to the head of the fibula in the transverse plane. 

#### 2.2.4. Test Protocol

Five healthy participants (3 male, 2 female, age 37 ± 12.6 years old, weight 78.28 ± 15.9 kg) completed the test. Each participant provided written, informed consent and the study was approved by the institutional ethics review board (REB: 21-12-022). The APAD was mounted onto a Saris M2 Smart Trainer™ (Saris, WI, USA) that provided resistance through magnetic impedance and controlled over Bluetooth™ by modulating the power of the resistance. The participant was asked to place the ball of their foot slightly ahead of the axis of rotation of the pedal to conform to recommended cycling practice [33]. A 5-min warm-up trial was performed, where the participant was allowed to pedal the APAD at 40 Watts resistance and preferred cadence. The actual pedaling task protocol consisted of 3 sequential trials, each having consecutive APAD active (A) and inactive (I) sessions, each lasting for 2 min (Figure 4). The pedaling cadence and trainer resistance were consistent within each trial, however, the APAD assistive system was active in session (A) for the crank angle of 90° to 270°, and inactive for the rest of the cycle labeled as session (I). A metronome played audible tones at twice the defined cyclic speed of the test to help participants stay at the target cadence. Upon request, participants were given a rest period between the two trials. 

#### 2.2.5. Measurements

**Assessment of kinematics of motion:** Assessment of kinematics of motion: To verify that the APAD did not significantly change the kinematics of the pedaling task, we computed the variation in pedaling cadence and knee lateral deviation in the transverse plane (Figure 5). The center point of the lateral-medial path of the knee joint (CP-LMP) within a cycle (marker points in Figure 5) was considered as the parameter describing the lateral position of the knee joint at each cycle.

**Assessment of asymmetric pedaling torque assistance:** The APAD performance in providing asymmetric torque assistance was evaluated by comparing the pedal force profile over a full cycle when APAD was active and inactive. Additionally, the significance of pedaling force reduction on muscle activity was studied using the EMG of VL and GM muscles. 

**Statistics:** A two-tailed *t*-test (α = 0.05) was conducted to evaluate the significance of parameters variation between two test conditions, including the area under the pedal force curve (AUC) of full cycles for sessions (A) and (I) to verify that the force reduction was significant. 

## 3. Results and Discussion

The CP-LMP of sessions (A) and (I) were not statistically different, indicating that the knee lateral-medial displacement was unaffected when the APAD became active. For both sessions (A) and (I), the crank angular velocity had about ±10 RPM real-time variation (“+” markers in Figure 6), which illustrates the participant’s attempt to adjust the pedaling cadence according to the metronome. In Trials 1 and 3, the average crank angular velocity over 3 s (solid line in Figure 6) had a small variation from its target velocity (dashed line in Figure 6), but this variation was more significant in Trial 2. This could be due to relatively higher angular velocity and cycling resistance in Trial 2, making it harder for participants to pedal at the target angular velocity. Additionally, muscle fatigue may have challenged the participant to maintain a consistent angular velocity in Trial 2. Nevertheless, all participants could maintain the target angular velocity and synchronize their pedaling speed with the metronome within an acceptable tolerance. Figure 5 and Figure 6 indicated that the kinematics of pedaling task was not significantly affected by the APAD torque assistance, and no abnormal changes were observed in the motion parameters when the APAD was active. This is important when the safety of the participant is concerned, particularly when APAD will be used for rehabilitation purposes.

Figure 7 shows the amount of the crank perpendicular force measured by strain gauges during a full cycle of the crank for Trials 1 to 3. The overlap of force profile in sessions (A) and (I) was less in Trial 3, indicating that the APAD assistive pedal torque was more effective, which was also noted by the participants during the tests. The AUC of the force profile was computed to quantify the pedal force variation in response to APAD torque assistant. The mean and standard deviation of AUC and its corresponding *p*-value for the target leg is given in Table 2. The AUC was reduced by 19% to 40% in session (A) compared to the corresponding session (I), depending on the pedaling power. The *p*-value was less than 0.001 for all trials, which confirms that a significant reduction in pedaling force occurred when APAD was activated at different cadences and intensities. 

The AUC parameter is correlated to the performed work for a revolution of the crank. Thus less muscle activity is expected in session (A) compared to the corresponding session (I) for the target leg. However, the significance level of muscle activity reduction should be studied because there is a possibility that the activity of the muscle of interest was not affected. The gastrocnemius (GM) and vastus lateralis (VL) muscles are among the major contributors to force production in the pedaling task. Figure 8 and Figure 9 illustrate the polar plot of the GM and VL muscle activities. The radius and angle of the polar plot represent the magnitude of EMG and the crank angle, as defined previously, respectively. The GM and VL’s range of active angles can be easily interpreted from the polar plots. The GM muscles of the left and right legs had more than 50% of their respective maximum activity during the crank angle range of 180–360° (±30° SD) (Figure 8). This range for the VL muscle was 90–240° (±10° SD) (Figure 9). This means that the VL muscle had a more significant contribution in generating pedaling torque during the downstroke (crank angle 90° to 270°) compared to the GM muscle, which was also reported in previous studies [34,35]. The EMG profile of GM and VL muscles of the non-target limb did not change between sessions (A) and (I) as much as the corresponding muscles of the target leg. This could be an indication that the APAD system reduced the muscle load of the target leg, without affecting the muscle load of the non-target leg, which was the objective of developing APAD. 

The APAD capability in providing asymmetric pedaling torque assistance allows for optimizing the intervention programs for patients with lower limb conditions. However, some limitations in the current design of the APAD should be addressed to make it a clinical tool. The strain gauge-based force sensor in the present study was not sensitive enough to measure crank axial force. Although the crank axial force does not contribute to the pedaling torque, it should be measured to determine the foot force, an essential parameter for musculoskeletal analysis. The other limitation of the present study relates to recruiting healthy subjects for the experiments, which was a safety consideration when testing the first prototype of APAD. The performance of APAD in reducing the pedaling force and muscle activity and its effect on the pain level will be studied in future human studies on patients with knee OA and ACL injuries. Finally, the open loop motor control system of APAD should be improved to a feedback loop control system, where the biomechanical responses, such as pedaling power, and physiological responses, such as heart beat rate, of the user are used to control the magnitude and duration of the motor power assistance. The next study will address those limitations to take the APAD one step closer to becoming a clinical human assistive device. 

## 4. Conclusions

The instruments designed and built for APAD were able to track crank angle and crank perpendicular force with acceptable accuracy. The results of the human study with healthy participants showed that APAD did retain the pedaling kinematics while the foot force and muscle force (kinetics) of the target leg were reduced. APAD showed that it could successfully assist the participant asymmetrically. However, the human study in the present work was limited to healthy participants. Thus, the next step is to evaluate the effectiveness of APAD as a rehabilitation device using human studies on patients with lower limb asymmetric function, such as post-stroke patients.

## Figures and Tables

**Figure 1 sensors-23-02846-f001:**
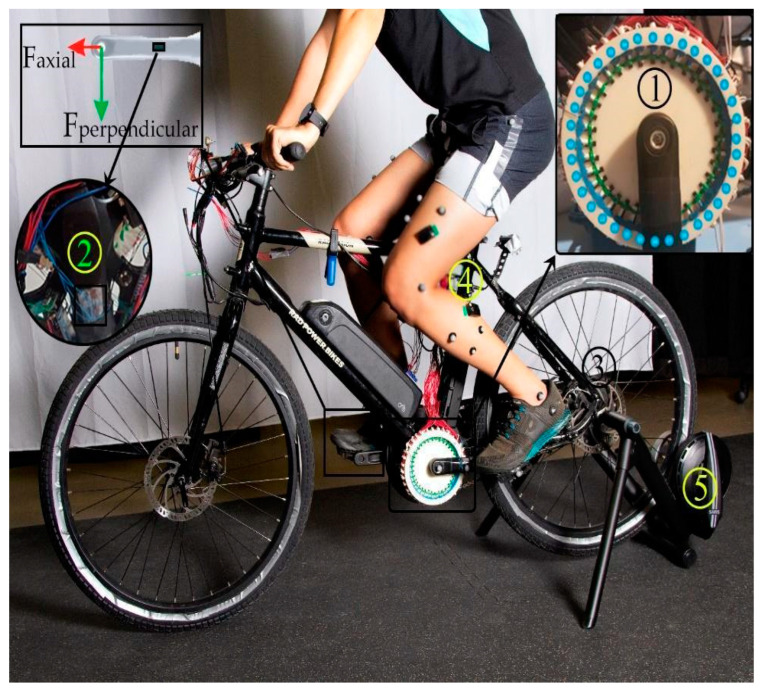
The APAD consisted of a crank position sensing system (1) and force sensor (2), BLDC rear hub motor (3), and a controller (4). The APAD was mounted on a trainer (5) for testing in the motion capture lab. The crank tracking unit consisted of 36 hall sensors on the 3D-printed fixture. The strain gauges-based force sensor measured the crank’s perpendicular force. The ID number of each hall sensor to the controller is pictured along the circumference of the unit.

**Figure 2 sensors-23-02846-f002:**
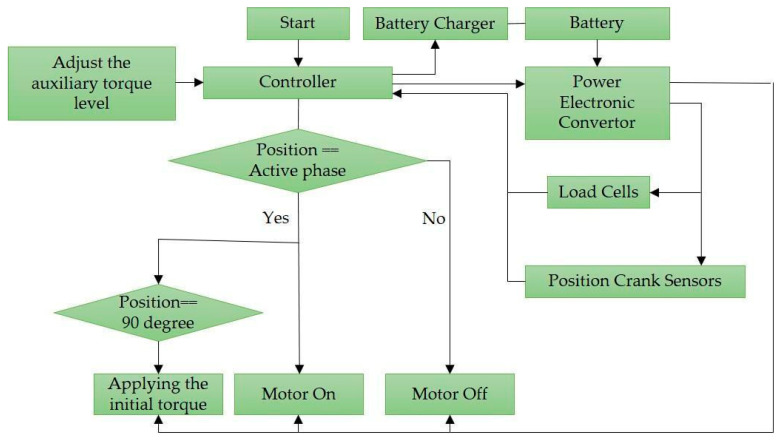
APAD motor controller flowchart.

**Figure 3 sensors-23-02846-f003:**
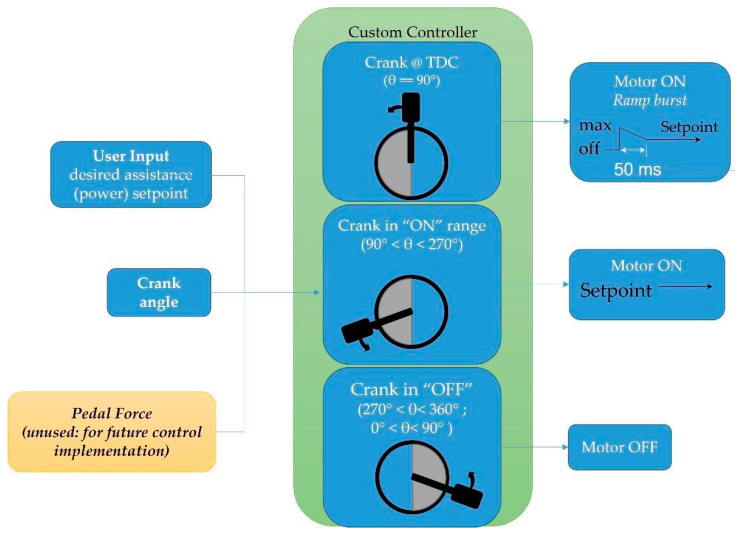
The custom motor control method implemented in the APAD provided assistive torque to the target leg.

**Figure 4 sensors-23-02846-f004:**
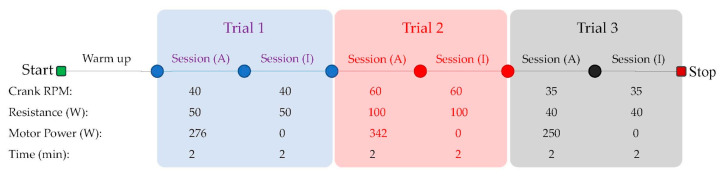
Experimental protocol at three different cadences and motor power assistance which occurred during Trials 1 to 3. Each trial consisted of two sessions of a pedaling task, where the APAD system was active in session (A) and inactive in session (I).

**Figure 5 sensors-23-02846-f005:**
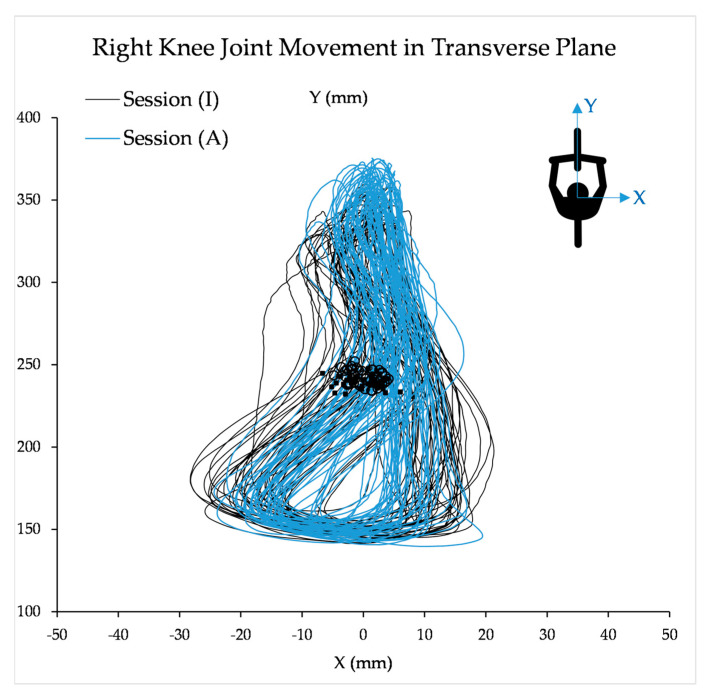
The lateral-medial movement of the target knee joint marker in the transverse plane was captured by the motion capture system, when APAD was inactive (I) and active (A). The square (■) and circle (○) markers represent the center point of the knee joint position during a full cycle for sessions (I) and (A) in Trial 1, respectively. The result is for a representative participant.

**Figure 6 sensors-23-02846-f006:**
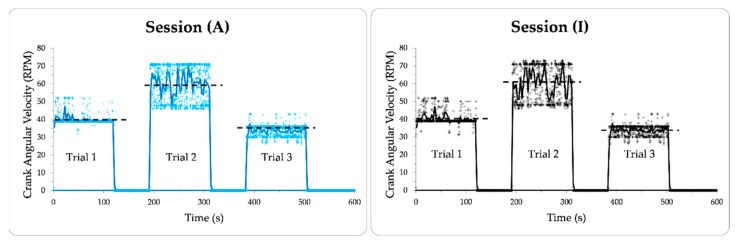
The distribution of real-time crank angular velocity (marker style “+”) and its average (solid line, window = 3 s) during sessions (A) and (I) through Trials 1 to 3. The target angular velocity is shown with the dashed line.

**Figure 7 sensors-23-02846-f007:**
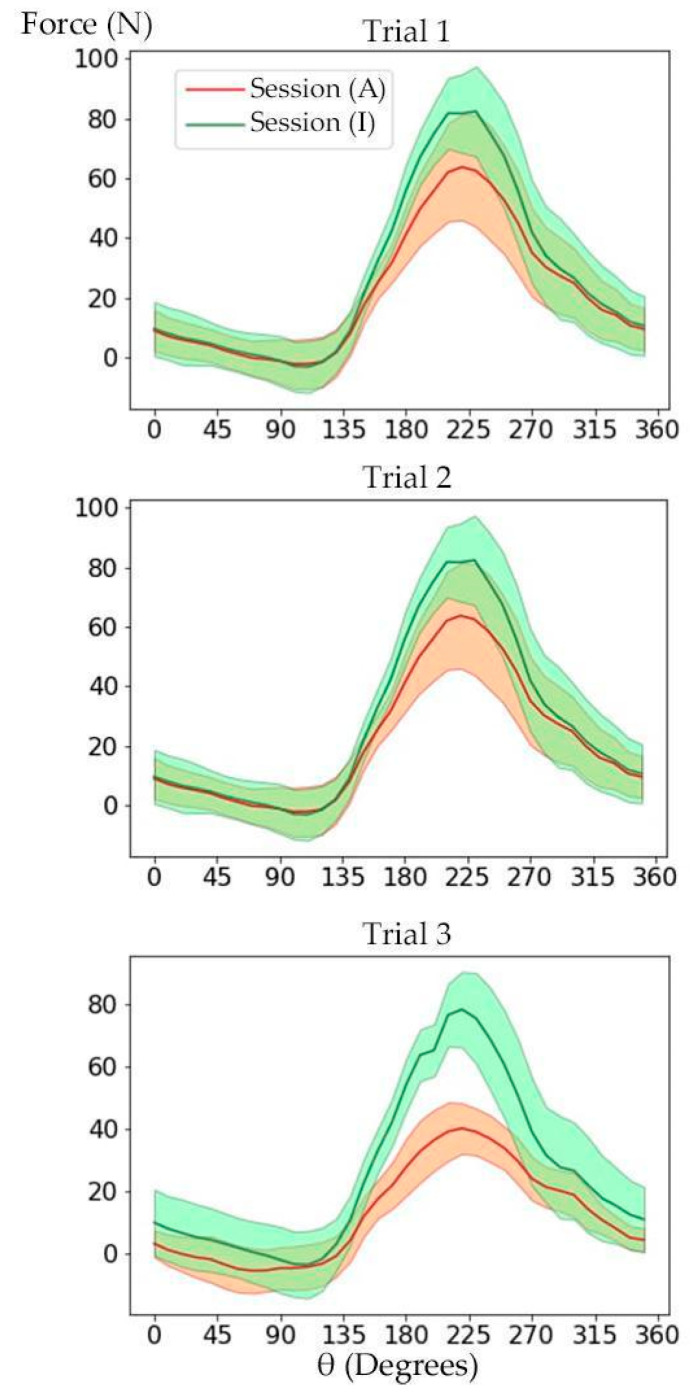
The perpendicular pedal force of the target leg was measured by the strain gauge system. The force curves over a crank revolution during sessions (A) and (I) were averaged. The highlighted regions show the standard deviation. The negative force represents the crank positions for which the crank perpendicular force creates a torque in the opposite direction of motion, performing a negative work. This happened mainly for the crank angle 270° to 360 + 90°, when the leg weight applied a negative torque.

**Figure 8 sensors-23-02846-f008:**
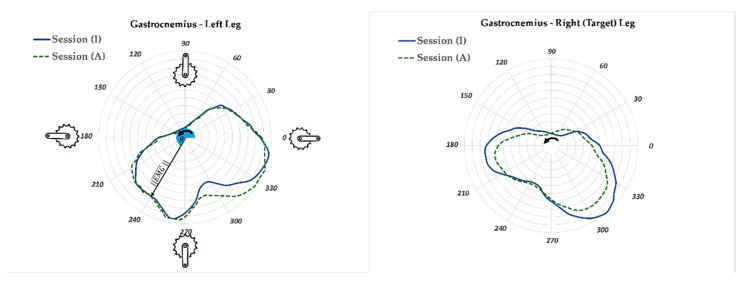
The Polar plot of gastrocnemius muscle activity of left and right legs for Trial 2 (60 RPM). The GM was active mostly when the crank angle varied from 180–360° (±30° SD). The gastrocnemius muscle activity was reduced when APAD was active for the target leg (right leg). The radius $\left|\left|\underline{EMG}\right|\right|$, and angle θ represent the magnitude of normalized EMG and the crank angle, respectively. The arrow shows the direction of rotation.

**Figure 9 sensors-23-02846-f009:**
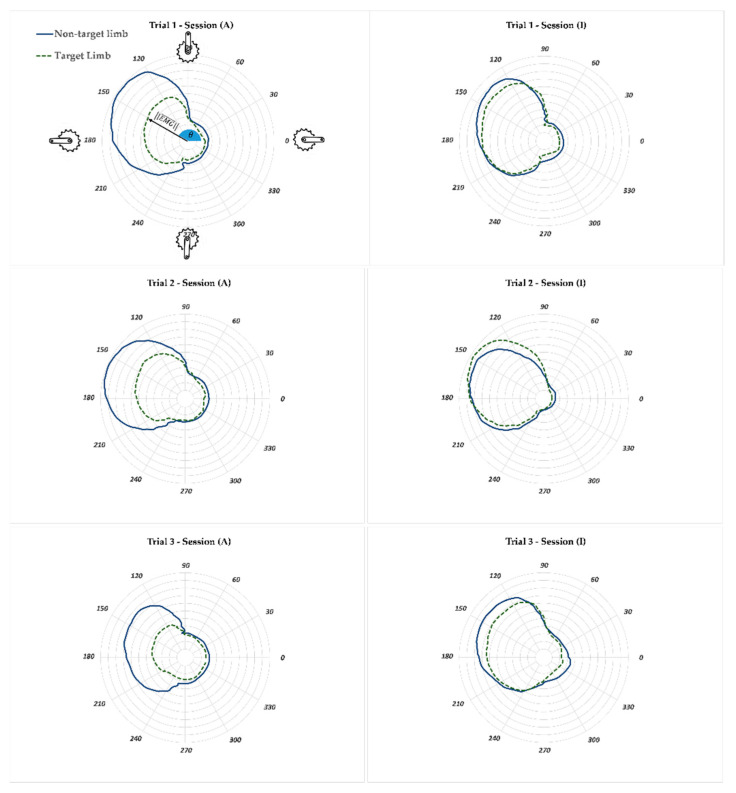
The polar plot of normalized VL muscle activity of target (**right**) and non-target (**left**) legs for Trials 1 to 3. The VL muscle was excited more than 50% of its respective maximum during the downstroke, i.e., crank angle 90–240° (±10° SD). During session (I), when the APAD was inactive, the left and right VL muscle activity profiles overlapped, indicating that the APAD did not assist the non-target leg. The radius $\left|\left|\underline{EMG}\right|\right|$, and angle θ represent the magnitude of normalized EMG and the crank angle, respectively.

**Table 1 sensors-23-02846-t001:** The main components of the APAD prototype.

Component	Type/Technology	Specification
Master Controller	Arduino Mega	54 I/O pin, 16 Analog pin
Slave Controller	Arduino Nano	Small board, based on the ATmega328
Crank Position Sensing System	Hall sensor (A3144E)	Weight: 1 gr, Digital Output Sensor
Force sensor	Strain gauge	Resistance: 349.8 ± 0.1 Ω, Sensitivity coefficient (gauge factor): 2.0–2.20
Actuator	Brushless DC motor	450 W BLDC rear-hub motor,
Communication device	Radio module (NRF24)	2.4 GHz band transceiver
Power Supply	Battery	48 Volts, 13 AH
Smart Trainer	Saris M2	±5% accuracy, Noise level: 69 decibels at 20 mph

**Table 2 sensors-23-02846-t002:** The AUC of the crank force of the target leg. The APAD was active during session (A), providing pedaling torque assistance to the target leg. Numbers inside parentheses are standard deviations. A *t*-test was conducted to compare values in sessions (A) and (I).

	AUC of Crank Perpendicular Force		
	Trial 1	Trial 2	Trial 3
Session (A)	2778 (407)	1828 (451)	1822 (442)
Session (I)	3521 (560)	2247 (453)	3938 (734)
*p*-value	<0.001	<0.001	<0.001

## Data Availability

Data is unavailable due to privacy or ethical restrictions.

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
