# Peer review of "An Adaptive Pedaling Assistive Device for Asymmetric Torque Assistant in Cycling"

_sensors, 2023, doi:10.3390/s23052846_

Round 1
Reviewer 1 Report
The paper presents some interesting results regarding the APAD and its application in the rehabilitation field. In any case, the overall treatment of the subject is not exhaustive and it is difficult to understand the proposed innovation. Fundamental sections such as Abstract and Conclusions are covered in a summary manner. The bibliography is mostly old and only rarely refers to reference journals for the sector.
I suggest a careful review of the article and a treatment more consistent with the level of the journal.
Author Response
Please refer to the attached file that contains our point-by-point responses to questions.

Reviewer 2 Report
Authors present a an Adaptive Pedaling Assistive Device for A symmetric Torque Assistant. Indeed, a new cycling ergometer was developed, and its function was validated using human tests. The developed cycling ergometer was equipped with sensors to record the kinetics and kinematics of pedaling, and electromechanical systems to provide assistive pedalling power only to the target leg.
This is an interesting work, In order to improve it, I suggest the following points :
- Abstract is not well written, it should be optimized illustrating the main contribution.
- In the introduction, the comparison with previous works is well precise highlighting the real contribution of this work. In addition, the motivation and background of wide practical use of the theoretic results presented are clearly emphasized to facilitate the readers.
- Experimental platform should be summarized in a table illustrating the main specifications of each component.
- Fig2: the main controller should be more explained.
- Section : 2.2.2. EMG Data Filtering: The acquisition system is not presented, the selection of the sampling frequency as well as the filter parameters is not justified. In addition, the choice of the RMS feature extraction approach is not discussed.
- The motion capture approach is also too reduced. it should be more explained.
- Results are well presented, however, the used method is too reduced and should be more detailed.
- The Conclusion should be rewritten by integrating the limitations and the perspective.
Concluding, the paper has potential to be appreciated by the readers and the above comment are formulated such that to enhance its impact.
Author Response

(The authors gave the same response as above.)

Round 2
Reviewer 1 Report
The manuscript has been sufficiently improved to warrant publication in Sensors
Reviewer 2 Report
The authors reacted properly to my pointed issues.